# Zero-inflated Poisson regression analysis of factors associated with under-five mortality in Ethiopia using 2019 Ethiopian mini demographic and health survey data

**Alemayehu Siffir Argawu** ⬤ *, **Gizachew Gobebo Mekebo**

Department of Statistics, Ambo University, Ambo, Oromia Region, Ethiopia

* alex089973@gmail.com

## Abstract

### Background

Remarkable reduction in global under-five mortality has been seen over the past two decades. However, Ethiopia is among the five countries which account for about half (49%) of all under-five mortality worldwide. This study aimed at identifying factors associated with under-five children mortality in Ethiopia using the 2019 Ethiopia mini demography and health survey data.

### Methods

The most recent national representative demography and health survey data of Ethiopia, 2019 Ethiopia mini demography and health survey data, were used for this study. Count data regression models were applied to identify the factors associated with under-five children mortality. Statistical significance was declared at P-value less than 0.05.

### Results

Zero-Inflated Poisson (ZIP) regression model was found to be the best model compared to other count regression models based on models comparison Criteria. The ZIP model revealed that decreased risk of under-five mortality was associated with mothers aged 25–34 years, unmarried mothers, mothers delivered in health facility, mothers used Pill/IUD, mothers who had larger number of children at home whereas increased risk of under-five mortality was associated with older mothers at their first births, mothers from rural areas, mothers travel for 1–30 min and >30 min to get drinking water, mothers used charcoal and wood, children with higher birth order and multiple births.

### Conclusions

In this study, place of residence, region, place of delivery, religion, age of mother, mother's age at first birth, marital status, birth order, birth type, current contraceptive type used, type of cooking fuel, time to get drinking water, and number of children at home were statistically

**Data Availability Statement:** The data used for the final analysis in this study are in 'Supporting Information'.

**Funding:** The authors received no specific funding for this work.

**Competing interests:** The authors have declared that no competing interests exist.

**Abbreviations:** AOR, Adjusted odds ratio; AIC, Akaike's Information Criterion; BIC, Bayesian information criterion; IRR, incidence rate ratio; DHS, Demographic and Heath survey; EDHS, Ethiopian Demographic and Heath survey; EMDHS, Ethiopian Mini Demographic and Heath survey; HNB, Hurdle Negative Binomial; HP, Hurdle Poisson; NB, Negative Binomial; SNNPR, Southern Nations, Nationalities, and Peoples' Region; U5, Under-five; U5M, Under-five mortality; WHO, World Health Organization; ZIP, Zero-Inflated Poisson; ZINB, Zero-Inflated Negative Binomial.

significant factors associated with under-five mortality in Ethiopia. Thus, the Ethiopian Ministry of Health and other concerned bodies are recommended to encourage mothers to deliver at health institutions, give awareness for mothers to use Pill/IUD contraceptive type, and facilitate rural areas to have electricity and drinking water near to homes so as to minimize the under-five mortality to achieve the sustainable development goal.

## Introduction

Remarkable reduction in global under-five mortality has been seen from 93 deaths per 1000 live births (12.83 million deaths) in 1990 to 38 deaths per 1000 live births (5.03 million deaths) in 2021 [1]. Despite this reduction, more than half (2.82 million deaths) of under-five deaths were in from sub-Saharan Africa [1]. The sustainable development goal targets even to reduce the global under-five death to at least 25 deaths per 1000 live births by 2030 [2].

About half (49%) of all under-five deaths occurred in five countries: Nigeria, India, Pakistan, the Democratic Republic of the Congo and Ethiopia in 2019 [3]. Children born in sub-Saharan Africa are at the highest risk of childhood death in the world with under-five mortality rate of 74 deaths per 1000 live births, which is 15 times higher than the risk for children in Europe and Northern America in 2021 [1].

In 2021, the under-five death rate in Ethiopia was 47 deaths per 1000 live births [1]. The Ethiopian recent demographic and health survey, 2019 EMDHS, report indicates that under-five death rate decreased to 59 deaths per 1000 live births in 2019 from 123 deaths per 1000 live births in [4].

Though the progress in reduction in under-five mortality has been made in Ethiopia, it seems inadequate to attain the Sustainable Development Goal of reducing to fewer than 25 deaths per 1000 live births by 2030. It is crucial to identifying factors associated with under-five mortality so as to inform public health policies and design strategies to accelerate the reduction of under-five mortality [5,6]. Many prior studies conducted to analyze the factors associated with under-five mortality in Ethiopia using the demographic and health survey data, did not include some important variables like type of cooking fuel, time to get source of water, age of household head, number of household members, relationship to household head, number of under-five children, and number of children at home [7–10]. This study, therefore, aimed at identifying factors associated with under-five mortality in Ethiopia using the 2019 Ethiopia mini demography and health survey data.

## Methods

### Study design, data source and population

The study was a retrospective design study, and the data source was the 2019 Ethiopia mini demography and health survey data. The census frame was a complete list of 149,093 enumeration areas (EAs), of which 35292 were in urban areas and 113801 in rural areas. In the first stage, 305 EAs (93 in urban areas and 212 in rural areas) were selected with probability proportional to EAs size and the household listing was carried out in each EA. In the second stage, 30 households per cluster were selected with equal probability selection [4].

### Sample in the study

A total of 8663 households were successfully interviewed with a response rate of 99%. In the interviewed households, 9012 women aged 15–49 were identified for individual interviews [4].

Finally, the researcher extracted 5535 number of weighted women with under-five children from the data.

## Study variables

**Response variable.** Response variable was total number of children died before celebrating fifth birthday per woman in her lifetime measured as count 0, 1, 2. . .

**Independent variables.** Independent variables were mother's age, mother's education level, mother's literateness, marital status, religion, mother's age at first birth, place of delivery, current contraceptive type, residence, region, number of women in the home, source of water, time to get source of water, toilet facility, age of household head, wealth index, number of household members, relationship to household head, type of cooking fuel, number of under-five children, number of children at home, birth order, birth type, and sex of child.

## Statistical data analysis methods

We applied the count regression models to analyze the data as the response variable, total number of children died before celebrating fifth birthday per woman in her lifetime measured as 0, 1, 2. . ., was the count data. The under-five mortality data experienced excess zeros characterized by over-dispersion and heteroscedasticity. The most popular distribution for modeling such data was the zero-inflated model and hurdle models. The over-dispersion has been explained as heterogeneity that has not been accounted for unobserved population which consists of several sub-populations in this case of Poisson type, but the sub-population membership is not observed in the sample. This excess variation may be occurred incorrect inference about parameter estimates, standard errors, tests, and confidence intervals. The Negative binomial model addresses the issue of over-dispersion by including a dispersion parameter to accommodate the unobserved heterogeneity in the count data. However, it cannot address the over-dispersion caused by an excessive number of zeros, in such case zero-inflated and Hurdle models are appropriate. Zero-inflated models mix a count component and a point mass at zero, allowing for over-dispersion [11,12].

The likelihood-ratio test is used to test the null hypothesis of no over-dispersion (i.e., the Poisson model is preferred) against the alternative hypothesis the over-dispersion parameter is different from zero (i.e., the data would be better fitted by the negative binomial regression). Furthermore, log likelihood, MSE, MAE, AIC and BIC were used to compare various candidate models, and the model with the smallest AIC and BIC value was considered as a better fit [13]. The data analysis was done by using SPSS 25, STATA 14, and R 4.1.0 versions software packages.

## Ethical consideration

Authors got permission from Demographic and Health Surveys (DHS) Program and downloaded data. As this data is publicly available and has no personal identifiers, Ethical approval was not necessary.

# Results

## Descriptive statistics

A total of 5535 women were included, of which 1277 (23.07%) women had lost at least one under-five children by death before celebrating fifth birthday whereas the remaining 4258 (76.93%) of the mothers had not lost their U5 children by death. This indicates zero outcome were large in number. The histograms are highly picked at the beginning (the zero values).

**Table 1. Number of under-five children deaths per mother, EMDHS 2019.**

| Number of deaths | Frequency | Percent |
|---|---|---|
| 0 | 4258 | 76.93 |
| 1 | 836 | 15.10 |
| 2 | 305 | 5.51 |
| 3 | 99 | 1.79 |
| 4 | 21 | 0.38 |
| 5 | 6 | 0.11 |
| 6 | 8 | 0.14 |
| 7 | 2 | 0.04 |
| Total | 5535 | 100 |
| Mean | 0.35 | |
| Variance | 0.57 | |

However, large number of under-five deaths per mother were observed less frequently. Additional screening of the number of child deaths showed that the variance (0.57) was greater than the mean (0.35) indicating over-dispersion. The analyses were summarized in Table 1.

## Models comparisons criteria

At the point when the significant wellspring of over-dispersion is a dominance of zero tallies, the subsequent over-dispersion cannot be modeled precisely with the negative binomial regression model. An elective path for demonstrating this kind of data is the zero-inflated Poisson or zero-inflated negative binomial regression model which considers the excess of zeroes. And, the overall models comparison was presented in Table 2. The minimum BIC was observed for the NB model, followed by Poisson and ZIP models. However, other validity indices of the model (maximum log likelihood and minimum MSE and MAE) favored for ZIP and ZINB models over all other models. But, the ZIP model is more preferable than ZINB by minimum AIC. In addition, the plot of observed minus predicted probability of the number of U5 deaths at each count was displayed in Fig 1. The line of difference between observed minus predicted probability of the number of U5 deaths was close to the reference zero line, showing the data is better fit of ZIP model than ZINB and other models.

## Mothers socio-demography determinants in the fitted model

Table 3 shows socio-demography determinants of under-five mortality in the ZIP regression model. According to the result of this study, the rate of non-zero U5 death for 25–34 years old

**Table 2. Overall models comparison by model fit characteristics.**

| Test statistics | Model | | | | | |
|---|---|---|---|---|---|---|
| | Poisson | NB | HP | HNB | ZIP | ZINB |
| Observed 0 value | 4258 | 4258 | 4258 | 4258 | 4258 | 4258 |
| Predicted 0 value | 4214 | 4250 | 4258 | 4258 | 4307 | 4307 |
| Log likelihood | -3343.9 | -3337.4 | -3154.8 | -3154.8 | -3115.9 | -3115.9 |
| AIC | 6799.9 | 6788.8 | 6533.6 | 6535.6 | **6455.8** | 6457.8 |
| BIC | 7170.6 | 7166.1 | 7274.8 | 7283.5 | 7197.2 | 7205.8 |
| MSE | 5.377 | 5.445 | 0.394 | 0.394 | 0.375 | 0.375 |
| MAE | 2.083 | 2.095 | 0.365 | 0.365 | 0.358 | 0.358 |

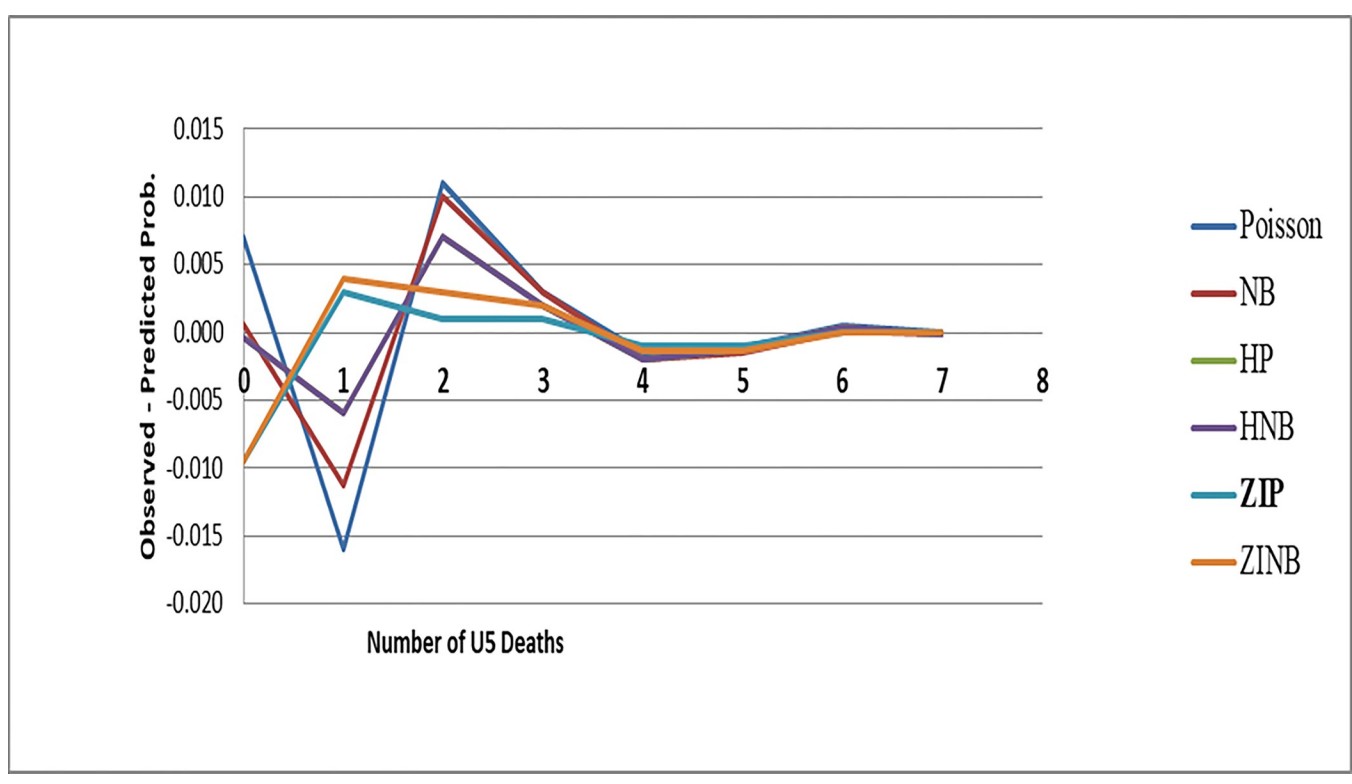

**Fig 1. Plots of observed minus predicted probability of number of U5 deaths for six models.**

mother decreased by 27% (IRR = 0.73, 95% CI: 0.58–0.89) as compared with 15–24 years old mother with keeping other variables held constant in the model. Associated with mother's marital status, the risk of U5M for unmarried mothers was 0.68 (IRR = 0.68, 95% CI: 0.5–0.86) times lower among U5 children born from married mothers. With regard to mother's religion, the risk of U5M for mothers who follow Muslim, protestant and other religions were 1.35 (IRR = 1.35, 95% CI: 1.18–1.51), 1.18 (IRR = 1.18, 95% CI: 1.00–1.35) and 1.69 (IRR = 1.69, 95% CI: 1.16–2.23) times more likely to die before age five as compared to mothers who follow Orthodox religion, respectively. Likened with mothers ages at 1st birth, the rates of U5 death for 15–24 and above 24 years old mothers increased by 24% (IRR = 1.24, 95% CI: 1.05–1.44) and 66% (IRR = 1.66, 95% CI: 1.3–2.03) as compared with below 15 years old mother, respectively. Based on place of delivery, mother who delivered in health facility had 0.59 (IRR = 0.59, 95% CI: 0.52–0.66) times less likely U5 death rate as compared with mother who delivered in home. Similarly, compared to mothers who used contraceptive type currently, the incidence rate of U5 death for mothers who used Pill/IUD was decreased by 36% (IRR = 0.64, 95% CI: 0.41–0.88) as compared with mothers who not used any contraceptive type before the survey. In the zero-inflated part, the estimated odds that the number of U5 death becomes zero for mothers who follow Muslim and protestant decreased by 87% (AOR = 0.13, 95% CI: 0.06–0.21) and 45% (AOR = 0.55, 95% CI: 0.18–0.68) as compared to mothers who follow Orthodox religion, respectively. The analyses were summarized in Table 3.

### Mother's household head related determinants in the fitted model

Table 4 displays that mother's household head related determinants of under-five mortality in the ZIP regression model. In the count part of the model, the risk of U5M among mothers

**Table 3. ZIP regression fitted model for the number of under-five children deaths by mothers' socio-demographic and related characteristics categories in Ethiopia, EMDHS 2019.**

| Count inflation model coefficients | | | | | | | | |
|---|---|---|---|---|---|---|---|---|
| Variable | Category | Estimate | SE | Z-value | P-value | IRR | 95% CI for IRR | |
| Intercept | Constant | -3.29 | 0.37 | -9.0 | 0.000 | 0.04 | 0.01 | 0.06 |
| Mother's education level (Ref: No education) | Primary | -0.11 | 0.06 | -1.96 | 0.05 | 0.90 | 0.79 | 1.00 |
| | Secondary | -0.08 | 0.12 | -0.66 | 0.512 | 0.93 | 0.72 | 1.14 |
| | Higher | -0.12 | 0.15 | -0.82 | 0.410 | 0.89 | 0.63 | 1.14 |
| Mother's age (Ref: 15–24 years) | 25–34 years | -0.31 | 0.11 | -2.87 | 0.004 | 0.73 | 0.58 | 0.89 |
| | 35–49 years | -0.05 | 0.13 | -0.4 | 0.692 | 0.95 | 0.71 | 1.19 |
| Marital status (Ref: Married) | Unmarried | -0.38 | 0.14 | -2.82 | 0.005 | 0.68 | 0.50 | 0.86 |
| Mother's religion (Ref: Orthodox) | Muslim | 0.3 | 0.06 | 4.7 | 0.000 | 1.35 | 1.18 | 1.51 |
| | Protestant | 0.16 | 0.08 | 2.16 | 0.031 | 1.18 | 1.00 | 1.35 |
| | Other | 0.53 | 0.16 | 3.26 | 0.001 | 1.69 | 1.16 | 2.23 |
| Literacy (Can't read sentence) | Read sentence | 0.02 | 0.08 | 0.26 | 0.795 | 1.02 | 0.87 | 1.17 |
| Mother's age at 1st birth (Ref: < 15 years) | 15–24 years | 0.22 | 0.08 | 2.7 | 0.007 | 1.24 | 1.05 | 1.44 |
| | Above 24 years | 0.51 | 0.11 | 4.59 | 0.000 | 1.66 | 1.30 | 2.03 |
| Place of delivery (Ref: Home) | Health facility | -0.53 | 0.06 | -8.73 | 0.000 | 0.59 | 0.52 | 0.66 |
| Current contraceptive used type (Ref: Not using) | Pill/IUD | -0.44 | 0.19 | -2.37 | 0.018 | 0.64 | 0.41 | 0.88 |
| | Injections | -0.09 | 0.08 | -1.1 | 0.270 | 0.91 | 0.76 | 1.06 |
| | Implants/Norplant | 0.16 | 0.1 | 1.68 | 0.093 | 1.18 | 0.95 | 1.4 |
| | Other | 0.13 | 0.18 | 0.70 | 0.487 | 1.14 | 0.73 | 1.54 |
| Zero inflation model coefficients | | | | | | | | |
| Variable | Category | Estimate | SE | Z-Value | P-value | AOR | 95% CI for AOR | |
| Intercept | Constant | 0.12 | 2.15 | 0.06 | 0.954 | 1.13 | -3.7 | 5.91 |
| Mother's educ. level (Ref: No education) | Primary | 0.05 | 0.28 | 0.16 | 0.871 | 1.05 | 0.47 | 1.63 |
| | Secondary | 0.44 | 0.49 | 0.9 | 0.368 | 1.55 | 0.07 | 3.04 |
| | Higher | 0.79 | 0.66 | 1.2 | 0.23 | 2.2 | -0.6 | 5.03 |
| Mother's age (Ref: 15–24 years) | 25–34 years | 0 | 0.42 | 0.01 | 0.995 | 1.00 | 0.18 | 1.83 |
| | 35–49 years | 0.59 | 0.57 | 1.03 | 0.302 | 1.8 | -0.2 | 3.83 |
| Marital status (Ref: Married) | Unmarried | 0.32 | 0.61 | 0.53 | 0.598 | 1.38 | -0.3 | 3.05 |
| Mother's religion (Ref: Orthodox) | Muslim | -2.02 | 0.29 | -6.84 | 0.000 | 0.13 | 0.06 | 0.21 |
| | Protestant | -0.6 | 0.12 | -2.24 | 0.000 | 0.55 | 0.18 | 0.68 |
| | Other | -1.59 | 0.83 | -1.91 | 0.057 | 0.20 | -0.1 | 0.54 |
| Literacy (Can't read sentence) | Read sentence | 0.16 | 0.37 | 0.42 | 0.676 | 1.17 | 0.31 | 2.03 |
| Age of mother at first birth (Ref: < 15 years) | 15–24 years | -0.64 | 0.22 | -2.99 | 0.003 | 0.53 | 0.30 | 0.75 |
| | Above 24 years | -1.91 | 0.52 | -3.66 | 0.000 | 0.15 | 0.00 | 0.3 |
| Place of delivery (Ref: Home) | Health facility | -11.93 | 6.52 | -4.12 | 0.125 | 0.00 | 0.00 | 0.00 |
| Current contraceptive type (Ref: Not using) | Pill/IUD | -2.4 | 1.52 | -1.58 | 0.114 | 0.09 | -0.2 | 0.36 |
| | Injections | -0.15 | 0.39 | -0.4 | 0.692 | 0.86 | 0.2 | 1.51 |
| | Implants/Norplant | 0.53 | 0.56 | 0.93 | 0.351 | 1.69 | -0.2 | 3.56 |
| | Other | 1.9 | 0.86 | 2.21 | 0.027 | 6.69 | -4.6 | 18 |

from rural area was increased by 27% (IRR = 1.27, 95% CI: 1.06–2.24) compared with urban area mothers. Mothers who lived in Addis Ababa city decreases the incidence of U5M by 62% (IRR = 0.38, 95% CI: 0.13–0.63) compared with Tigray region. The risk of U5M was increased by 78%, 64%, 51% and 35% for mothers who lived in Afar, Somali, Oromia and Benishangul regions compared with Tigray region, respectively. The incidence rate of U5 death was increased by 62% (IRR = 1.62, 95% CI: 1.27–1.98) and 82% (IRR = 1.82, 95% CI: 1.42–2.22) for

**Table 4.** ZIP regression fitted model for the number of under-five children deaths by mothers' household related characteristics categories in Ethiopia, EMDHS, 2019.

| Count inflation model coefficients | | | | | | | | |
|---|---|---|---|---|---|---|---|---|
| Variable | Category | Estimate | SE | Z-value | P-value | IRR | 95% CI for IRR | |
| Residence (Ref: Urban) | Rural | 0.24 | 0.08 | 1.68 | 0.000 | 1.27 | 1.06 | 2.24 |
| Region (Ref: Tigray) | Afar | 0.58 | 0.15 | 3.85 | 0.000 | 1.78 | 1.26 | 2.31 |
| | Amhara | 0.23 | 0.15 | 1.55 | 0.122 | 1.26 | 0.89 | 1.62 |
| | Oromia | 0.41 | 0.14 | 2.96 | 0.003 | 1.51 | 1.10 | 1.93 |
| | Somali | 0.50 | 0.15 | 3.34 | 0.001 | 1.64 | 1.16 | 2.13 |
| | Benishangul | 0.30 | 0.15 | 2.00 | 0.045 | 1.35 | 0.95 | 1.75 |
| | SNNPR | -0.21 | 0.14 | -1.49 | 0.137 | 0.81 | 0.58 | 1.04 |
| | Gambela | 0.30 | 0.15 | 1.94 | 0.053 | 1.34 | 0.94 | 1.74 |
| | Harari | -0.03 | 0.16 | -0.18 | 0.856 | 0.97 | 0.68 | 1.27 |
| | Dire Dawa | -0.28 | 0.19 | -1.42 | 0.156 | 0.76 | 0.47 | 1.05 |
| | Addis Ababa | -0.97 | 0.34 | -2.84 | 0.005 | 0.38 | 0.13 | 0.63 |
| Source of water (Ref: Unimproved) | Improved | 0.06 | 0.06 | 1.01 | 0.313 | 1.06 | 0.94 | 1.17 |
| Time to get water (Ref: 0 Minutes) | ≤ 30 minutes | 0.48 | 0.11 | 4.32 | 0.000 | 1.62 | 1.27 | 1.98 |
| | > 30 minutes | 0.60 | 0.11 | 5.35 | 0.000 | 1.82 | 1.42 | 2.22 |
| Types of toilet facility (Ref: Unimproved) | Improved | -0.13 | 0.09 | -1.51 | 0.130 | 0.88 | 0.73 | 1.03 |
| Age of household head (Ref: 15–24) | 25–34 | 0.60 | 0.20 | 3.09 | 0.002 | 1.83 | 1.13 | 2.53 |
| | 35–44 | 0.55 | 0.20 | 2.71 | 0.007 | 1.73 | 1.04 | 2.41 |
| | Above 44 | 0.45 | 0.20 | 2.21 | 0.027 | 1.57 | 0.94 | 2.20 |
| Wealth index (Ref: Poor) | Medium | -0.08 | 0.08 | -1.03 | 0.301 | 0.92 | 0.78 | 1.06 |
| | Rich | -0.17 | 0.08 | -2.09 | 0.036 | 0.84 | 0.71 | 0.98 |
| Number of household members (Ref:1–3) | 4–6 | -0.20 | 0.15 | -1.31 | 0.190 | 0.82 | 0.58 | 1.06 |
| | Above 6 | -0.59 | 0.17 | -3.55 | 0.000 | 0.55 | 0.37 | 0.73 |
| Number of women in household (Ref: One) | Above one | 0.09 | 0.20 | 0.47 | 0.639 | 1.10 | 0.67 | 1.52 |
| Relationship to household head (Ref: Head) | Wife/husband | 0.03 | 0.09 | 0.34 | 0.733 | 1.03 | 0.85 | 1.21 |
| | Other | 0.50 | 0.14 | 3.59 | 0.000 | 1.64 | 1.20 | 2.08 |
| Type of cooking fuel (Ref: Electricity) | Charcoal | 0.62 | 0.24 | 2.63 | 0.009 | 1.86 | 1.00 | 2.71 |
| | Wood | 0.50 | 0.23 | 2.13 | 0.033 | 1.64 | 0.89 | 2.39 |
| | Other | 0.76 | 0.25 | 2.98 | 0.003 | 2.14 | 1.07 | 3.20 |
| **Zero inflation model coefficients** | | | | | | | | |
| Variable | Category | Estimate | SE | Z-value | P-value | AOR | 95% CI for AOR | |
| Residence (Ref: Urban.) | Rural | 0.45 | 0.31 | 2.71 | 0.000 | 1.32 | 1.12 | 2.48 |
| Region (Ref: Tigray) | Afar | 3.17 | 0.72 | 4.41 | 0.000 | 23.70 | -9.68 | 57.09 |
| | Amhara | 1.57 | 0.73 | 2.15 | 0.032 | 4.78 | -2.05 | 11.62 |
| | Oromia | 0.83 | 0.64 | 1.29 | 0.196 | 2.30 | -0.61 | 5.21 |
| | Somali | 1.65 | 0.66 | 2.49 | 0.013 | 5.19 | -1.54 | 11.92 |
| | Benishangul | 0.05 | 0.70 | 0.07 | 0.947 | 1.05 | -0.39 | 2.49 |
| | SNNPR | -1.77 | 0.85 | -2.09 | 0.037 | 0.17 | -0.11 | 0.45 |
| | Gambela | -0.53 | 0.70 | -0.75 | 0.452 | 0.59 | -0.22 | 1.40 |
| | Harari | -12.21 | 63.90 | -0.19 | 0.848 | 0.00 | 0.00 | 0.00 |
| | Dire Dawa | -22.05 | 859.05 | -0.03 | 0.980 | 0.00 | 0.00 | 0.00 |
| | Addis Ababa | -2.91 | 3.73 | -0.78 | 0.436 | 0.05 | -0.35 | 0.45 |
| Source of drinking water (Ref: Unimproved) | Improved | 0.16 | 0.25 | 0.65 | 0.517 | 1.18 | 0.60 | 1.76 |
| Time to get water source (Ref: 0 Minute) | ≤ 30 minutes | 0.18 | 0.17 | 1.09 | 0.276 | 1.20 | 0.81 | 1.59 |
| | > 30 min. | -0.14 | 0.12 | 0.45 | 0.210 | 0.87 | 0.63 | 1.11 |
| Toilet facility (Ref: Unimproved) | Improved | -0.51 | 0.31 | -1.68 | 0.093 | 0.60 | 0.24 | 0.96 |

*(Continued)*

**Table 4.** (Continued)

| | | | | | | | | |
|---|---|---|---|---|---|---|---|---|
| Age of household head (Ref: 15–24) | 25–34 | -0.06 | 0.52 | -0.12 | 0.908 | 0.94 | -0.02 | 1.90 |
| | 35–44 | 0.16 | 0.69 | 0.23 | 0.821 | 1.17 | -0.40 | 2.74 |
| | Above 44 | -0.30 | 0.70 | -0.43 | 0.670 | 0.74 | -0.28 | 1.76 |
| Wealth index (Ref: Poor) | Medium | -0.21 | 0.38 | -0.56 | 0.574 | 0.81 | 0.21 | 1.41 |
| | Rich | 0.44 | 0.42 | 1.05 | 0.295 | 1.55 | 0.28 | 2.82 |
| Number of household members (Ref:1–3) | 4–6 | 0.64 | 0.62 | 1.04 | 0.297 | 1.90 | -0.39 | 4.19 |
| | Above 6 | 0.71 | 0.63 | 1.12 | 0.263 | 2.03 | -0.48 | 4.54 |
| Number of women in household (Ref: One) | Above one | 3.90 | 4.28 | 0.91 | 0.363 | 49.24 | -364.1 | 462.2 |
| Relationship to household head (Ref: Head) | Wife/husband | -0.11 | 0.36 | -0.30 | 0.764 | 0.90 | 0.27 | 1.53 |
| | Other | 0.45 | 0.44 | 1.03 | 0.302 | 1.57 | 0.22 | 2.93 |
| Type of cooking fuel (Ref: Electricity) | Charcoal | 1.66 | 1.88 | 0.88 | 0.377 | 5.24 | -14.1 | 24.51 |
| | Wood | 2.97 | 1.93 | 1.53 | 0.125 | 19.41 | -54.0 | 92.95 |
| | Other | 1.81 | 2.03 | 0.89 | 0.372 | 6.11 | -18.2 | 30.36 |

mothers who traveled for ≤30 minutes and >30 minutes to obtain drinking water as compared with mothers who obtained drinking water near to their homes (or took 0 minute), respectively. Likewise, the incidence rate of U5 death was increased by 83%, 73% and 57% for mothers whose household heads ages are 25–34, 35–44, and above 44 years old as compared with 15–24 years old household head, respectively. The richest mothers had lower U5M rate (IRR = 0.84, 95% CI: 0.71–0.98) compared with poor mothers. Mothers from above six household members had lower U5Mrate (IRR = 0.55, 95% CI: 0.37–0.73) compared with mothers from 1–3 household members. Based on mothers' cooking fuel types, incidence of U5Mrate was increased by 86%, 64% and 114% for mothers who used charcoal, wood and other cooking fuel types compared with mothers used electricity fuel type, respectively. The analyses were summarized in Table 4.

## Child Related determinants in the fitted model

Table 5 illustrates that child related determinants of under-five mortality in the ZIP model. In the count part of the model, the incidence of U5Mrate was decreased by 63% (IRR = 0.37, 95% CI: 0.18–0.57) for the number of 5 and under children in the household is above three compared with only one child in the household. Likewise, the incidences of U5M rates were decreased by 49% (IRR = 0.51, 95% CI: 0.41–0.61) and 73% (IRR = 0.27, 95% CI: 0.20–0.33) for the number of U5 children in the homes are three to five and above five compared with two or less children in home, respectively. As birth order increases the U5Malso increases. The death rates of U5 whose orders of births are 2nd–3rd, 4th–5th, and above 5th were 3.91, 13.14, and 38.17 times higher compared with 1st birth order child, respectively. On the other hand, multiple birth type of child was 50% increase the risk of U5M (IRR = 1.5, 95% CI: 1.19–1.81) compared with a single birth type. In the zero-inflated part, the estimated odds that the number of zero U5 death for two children (5 and under) in the household was 4.09 time higher (AOR = 4.09, 95% CI: 1.41–6.77) compared with only one child in the household. The analyses were summarized in Table 5.

## Discussion

This study aimed, mainly, at identifying factors associated with under-five mortality in Ethiopia using 2019 EMDHS data. It was revealed that 25–34 years old mother had reduced the risk of under-five mortality as compared with 15–24 years old mother. This shows that a younger

**Table 5. Zero inflated Poisson regression fitted model for the number of under-five children deaths by children characteristics categories in Ethiopia, EMDHS 2019.**

**Count inflation model coefficients**

| Variable | Category | Estimate | SE | Z-value | P-value | IRR | 95% CI for IRR | |
|---|---|---|---|---|---|---|---|---|
| Number of 5 and under children in household (Ref: One) | Two | -0.04 | 0.06 | -0.71 | 0.478 | 0.96 | 0.84 | 1.07 |
| | Three | -0.35 | 0.21 | -1.64 | 0.102 | 0.71 | 0.41 | 1.00 |
| | Above 3 | -0.99 | 0.27 | -3.65 | 0.000 | 0.37 | 0.18 | 0.57 |
| Number of children at home (Ref: ≤ 2) | 3–5 | -0.67 | 0.10 | -6.73 | 0.000 | 0.51 | 0.41 | 0.61 |
| | 6 and more | -1.32 | 0.12 | -10.72 | 0.000 | 0.27 | 0.20 | 0.33 |
| Birth order number (Ref: First) | 2nd/3rd | 1.36 | 0.16 | 8.42 | 0.000 | 3.91 | 2.67 | 5.15 |
| | 4th/5th | 2.58 | 0.18 | 14.24 | 0.000 | 13.14 | 8.48 | 17.81 |
| | Above 5th | 3.64 | 0.19 | 19.24 | 0.000 | 38.17 | 24.0 | 52.33 |
| Sex of child (Ref: Male) | Female | -0.14 | 0.09 | -1.47 | 0.143 | 0.87 | 0.71 | 1.03 |
| Birth type (Ref: Single) | Multiple | 0.40 | 0.11 | 3.80 | 0.000 | 1.50 | 1.19 | 1.81 |

**Zero inflation model coefficients**

| Variable | Category | Estimate | SE | Z-value | P-value | AOR | 95% CI for AOR | |
|---|---|---|---|---|---|---|---|---|
| Number of 5 and under children in household (Ref: One) | Two | 1.41 | 0.33 | 4.22 | 0.000 | 4.09 | 1.41 | 6.77 |
| | Three | -3.20 | 4.33 | -0.74 | 0.460 | 0.04 | -0.3 | 0.39 |
| | Above 3 | -9.09 | 5.14 | -1.77 | 0.077 | 0.00 | 0.00 | 0.00 |
| Number of children at home (Ref: ≤ 2) | 3–5 | 0.22 | 0.39 | 0.57 | 0.570 | 1.24 | 0.30 | 2.19 |
| | Above 5 | 1.96 | 0.61 | 3.20 | 0.001 | 7.09 | -1.4 | 15.60 |
| Birth order number (Ref: First) | 2nd/3rd | -0.48 | 0.56 | -0.85 | 0.393 | 0.62 | -0.1 | 1.30 |
| | 4th/5th | -1.95 | 0.70 | -2.79 | 0.005 | 0.14 | -0.1 | 0.34 |
| | Above 5th | -4.56 | 0.79 | -5.79 | 0.000 | 0.01 | 0.00 | 0.03 |
| Sex of child (Ref: Male) | Female | -2.42 | 0.69 | -3.51 | 0.326 | 0.09 | 0.02 | 1.07 |
| Birth type (Ref: Single) | Multiple | -1.79 | 0.77 | -2.32 | 0.020 | 0.17 | -0.1 | 0.42 |

aged mother face higher child mortality risk. This finding is consistent with other studies conducted in different countries including Ethiopia [7–10], Kenya [14], Nigeria [15–17], Columbia [18], Pakistan [19], Bangladesh [20,21], Bolivia [22], and India [23]. This might be due to that younger mothers may not be socially and psychologically mature enough to deal with the requirements of infant and child care, or they may lack the domestic decision-making authority as compared with older mothers [20]. But, the finding is inconsistent with available literature that points to the fact that maternal age is a strong predictor of child survival [21,23–26].

The study also revealed that the incidence rate of under-five children death among children whose mothers' ages at first birth were 15–24 years and greater than 24 years were significantly more than among children whose mother's age at first birth is less than 15 years mothers. This finding agrees with result of [24,26]. But, it is contradicted with other related finding [27].

As presented in this finding, unmarried women had less likely under-five children deaths than the married counterpart. This finding is consistent with study conducted in sub-Saharan Africa countries [28]. However, this finding is inconsistent with other studies findings [15,23,28,29].

This study found that children of Muslim, protestant and other religion followed mothers were having higher risk of dying before the age of five years compared to children whose mothers followed orthodox religion. This might be partly due to the fact that Muslim women tend to face oppositions regarding the use of contraceptive methods from their husbands [30]. This might also be because that the Islam religion particularly forbids using contraception to restrict the overall size of families [31].

Place of delivery was also found to be another significant factor associated with the under-five mortality. Children born in a healthcare facility were at lower risk than those born at home. This finding agrees with studies [10,29,31,32]. This might be due to the proper health care and attention these facilities provide them during and after delivery.

Findings of this study indicated that less mortality rate of children's dying before age of five was associated with mothers using contraceptive type. Thus, children from mothers used pill/IUD contraceptive type is significantly less than children from mothers not used any contraceptive type. One of the important possible reasons for lower under-five mortality might be the longer birth intervals among the users. Similarly, this result is agreed with other previous findings [8,15,29].

The incidence rate ratio for the place of residence reveals that the chance of under-five mortality is higher in the rural area than in urban area. The mothers living in rural areas had significantly higher mortality rate than mothers living in urban areas. Several researchers found similar results showing children from rural area had high mortality rate than urban area [9,14,27,28,32–35]. The possible reason could be that urban areas are connected with quality health care services, good education and employment opportunities for mothers, implying a lower experience in children death.

Under-five mortality in Ethiopia was clearly varied between regions of the country. It seems likely that the socioeconomic and health factors variation exist between all regions and Tigray region had less under-five mortality rate as compared to other regions except Addis Ababa city. And, Addis Ababa city had a lower U5M and higher in other four regions compared with the Tigray region. This finding is in agreement with other studies which found that there were regional variation for infant and under-five mortality in Ethiopia [5,7,9,27], Kenya [36], Nigeria [15,16,37], Mozambique [38], Ghana [23,39], India [40], and Bangladesh [24] showed that regional variation significantly affects U5M. These differences might be because of difference in basic infrastructure distribution like health coverage and regional variations in economic development. Therefore, this finding demonstrates that providing an equal and fair distribution of wealth and health service coverage will tailor the variation observed among different regions of Ethiopia. However, one study showed that region was not significantly related with under-five mortality in Ethiopia [41].

Wealth index was another important differential factor of under-five mortality. The richest women had significant reduction in under-five mortality with compared to the poorest women. This result agrees with various findings [14,16,24,26,35,40,42–44]. Whereas, it is not agreed with a study conducted in Nigeria [29]. In Ethiopia, some previous related studies showed that the variable wealth index was not significantly associated with under-five mortality [31,41,45,46].

Birth order was another factor associated with under-five mortality. It had a parallel association with U5M. This study shows that the incidence rate of U5M increased with increased in birth order of the child. This is consistent with other previous studies [4,47,48]. Possible mechanism for this is that as birth order increases, intra-familiar competition for foods and other limited resources essential for child's need will be increased. Moreover, children are more prone to receive most impacts of it. Also as birth order increases level of child care reduces since the mother will have more children to care. But, others findings contradicted this idea [16,38,49].

Moreover, the study found that the under-five mortality was significantly associated with the age of household head, number of household members, time to get the source of drinking water, type of cooking fuel, number of children at home, and birth type. Higher U5M rate was associated with older household head. This finding is consistent with the study [50]. The higher U5M rate was associated with longer time to get the source of drinking water. This is

finding is consistent with the previous studies [43,50]. The higher U5M rate was also associated with cooking fuel type of charcoal, wood and other cooking fuel types other than electricity fuel type. This is consistent with prior studies [28,33,43,51]. Higher U5M rate was also associated with children of multiple births, which is consistent with the studies [8–10,32,39,48,52,53]. Lower U5M rate was associated with larger family size. This finding is consistent with studies [7,49]. The lower U5M rate was also associated with larger number of children at home. This is consistent with studies [15,54].

## Strength and limitations of the study

The main strength of this study was that it was analyzed using the nationally representative data with a large sample size. Additionally, it was based on count data regression models comparison to identify factors associated with under-five mortality. Moreover, since it is based on the national survey data the study has the potential to give insight for ministry of health, policy-makers, and other concerned bodies to design appropriate intervention strategies both at national and regional levels. The main limitation of this study that the EMDHS is mostly based on respondents' self-report and might have the possibility of recall bias. In addition, variables such as weight at birth [48,55,56], anemia status of mother [56–58], breastfeeding status of child [7,48,59], diarrhea [49,60,61], desire of pregnancy [62–65], employment status of mother [49,66,67], educational level of father [68–70] and maternal HIV status [71–73] were not included in the study due to large number of missing values/unavailability in the dataset.

## Conclusions

The ZIP regression model was found to be the best to fit the data and revealed that mothers' ages of 25–34 years old, unmarried mothers, mothers followed Muslim, protestant, and other religions, mothers' ages of 15–24, and above 24 years old at first birth, mothers delivered in health facility, mothers used Pill/IUD, mothers from rural area, mothers from Afar, Somali, Benishangul, Oromia regions and Addis Ababa city, mothers traveled for 1–30 minutes, and >30 minutes to obtain drinking water, and others were statistically significant determinants of U5M. Moreover, mothers traveled for long hours to obtain drinking water, mothers from Afar, Somali, Oromia and Benishangul regions, mothers from rural area, mothers delivered in homes, mothers used charcoal and wood cooking fuels, children of $2^{nd}$ and above birth orders, and multiple born children were associated with high incidence of U5M. Thus, the Ethiopian Ministry of Health and other concerned bodies are recommended to encourage mothers to deliver at health institutions, give awareness for mothers to use Pill/IUD contraceptive type, and facilitate rural areas to have electricity and drinking water near to homes so as to minimize the under-five mortality to achieve the sustainable development goal.

## Supporting information

**S1 Data.**
(SAV)

## Acknowledgments

The authors would like to thank the DHS Program for providing the data for the study.

## Author Contributions

**Conceptualization:** Alemayehu Siffir Argawu.

**Data curation:** Alemayehu Siffir Argawu, Gizachew Gobebo Mekebo.

**Formal analysis:** Alemayehu Siffir Argawu, Gizachew Gobebo Mekebo.

**Funding acquisition:** Alemayehu Siffir Argawu.

**Investigation:** Alemayehu Siffir Argawu, Gizachew Gobebo Mekebo.

**Methodology:** Alemayehu Siffir Argawu, Gizachew Gobebo Mekebo.

**Resources:** Alemayehu Siffir Argawu.

**Software:** Alemayehu Siffir Argawu, Gizachew Gobebo Mekebo.

**Supervision:** Alemayehu Siffir Argawu.

**Validation:** Alemayehu Siffir Argawu, Gizachew Gobebo Mekebo.

**Visualization:** Alemayehu Siffir Argawu, Gizachew Gobebo Mekebo.

**Writing – original draft:** Alemayehu Siffir Argawu.

**Writing – review & editing:** Alemayehu Siffir Argawu, Gizachew Gobebo Mekebo.

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
