## [Decision Letter · Decision Letter 0]

15 Jun 2023

PONE-D-22-20230Risk Factors of Under-Five Mortality Using Count Data Regression Models in Ethiopia, 2021PLOS ONE

Dear Dr. Argawu,

Thank you for submitting your manuscript to PLOS ONE. After careful consideration, we feel that it has merit but does not fully meet PLOS ONE’s publication criteria as it currently stands. Therefore, we invite you to submit a revised version of the manuscript that addresses the points raised during the review process.

Please note that we have only been able to secure a single reviewer to assess your manuscript. We are issuing a decision on your manuscript at this point to prevent further delays in the evaluation of your manuscript. Please be aware that the editor who handles your revised manuscript might find it necessary to invite additional reviewers to assess this work once the revised manuscript is submitted. However, we will aim to proceed on the basis of this single review if possible. 

We look forward to receiving your revised manuscript.

Kind regards,

Jianhong Zhou

Staff Editor

PLOS ONE

4. PLOS requires an ORCID iD for the corresponding author in Editorial Manager on papers submitted after December 6th, 2016. Please ensure that you have an ORCID iD and that it is validated in Editorial Manager. To do this, go to ‘Update my Information’ (in the upper left-hand corner of the main menu), and click on the Fetch/Validate link next to the ORCID field. This will take you to the ORCID site and allow you to create a new iD or authenticate a pre-existing iD in Editorial Manager. Please see the following video for instructions on linking an ORCID iD to your Editorial Manager account: https://www.youtube.com/watch?v=_xcclfuvtxQ.

Reviewers' comments:

Reviewer's Responses to Questions

**Comments to the Author**

1. Is the manuscript technically sound, and do the data support the conclusions?

Reviewer #1: Yes

2. Has the statistical analysis been performed appropriately and rigorously? 

Reviewer #1: No

3. Have the authors made all data underlying the findings in their manuscript fully available?

Reviewer #1: Yes

4. Is the manuscript presented in an intelligible fashion and written in standard English?

Reviewer #1: Yes

5. Review Comments to the Author

Reviewer #1: 1. Abstract: The result written in the abstract section is need to re-write.

2. Variable in study not been define clearly

3. I am skeptical about the method used in the study; survival regression can be used in spite of Zero-inflated Poisson regression.

4. Justifications of the methodology used has to be added.

5. what is newness in the study? and how it different from the existing studies form the same dataset.

6. PLOS authors have the option to publish the peer review history of their article (what does this mean?). If published, this will include your full peer review and any attached files.

Reviewer #1: No

---

## [Author Response · Author response to Decision Letter 0]

2 Aug 2023

Authors’ response to Editor and reviewers 

Dear Editor, 

Thank you so much for allowing resubmission of our manuscript with an opportunity to address the reviewers’ comments that could help to enhance the quality of the manuscript.

We would also like to express our gratitude to the anonymous reviewer for carefully reviewing the manuscript and for many thoughtful comments, which can enhance the readability and quality of this manuscript. 

We are uploading (i) our point-by-point response to the Editor’s and reviewer’s comments, and (ii) updated manuscript with changes being highlighted by red colour.

Best regards,

Alemayehu Siffir Argawu, the corresponding author of the manuscript

Author’s response for Academic editor comments 

 Authors’ Response: Thank you so much for the comment. The manuscript has been updated as per the PLOS ONE style requirements. 

Authors’ response: The data we drawn, for the analysis of this study, is publically open secondary data which is available on the DHS website (http:// dhsprogram.com). We got permission from Demographic and Health Surveys (DHS) Program and downloaded data. As this data is publicly available and has no personal identifiers, Ethical approval was not necessary. 

Authors’ Response: All data used for the final analysis for this study are available from corresponding author. We modified the statement in the submission system and uploaded the data.

Authors’ Response: We modified it in the Editorial system.

4. PLOS requires an ORCID iD for the corresponding author in Editorial Manager on papers submitted after December 6th, 2016. Please ensure that you have an ORCID iD and that it is validated in Editorial Manager. To do this, go to ‘Update my Information’ (in the upper left-hand corner of the main menu), and click on the Fetch/Validate link next to the ORCID field. This will take you to the ORCID site and allow you to create a new iD or authenticate a pre-existing iD in Editorial Manager. Please see the following video for instructions on linking an ORCID iD to your Editorial Manager account: https://www.youtube.com/watch?v=_xcclfuvtxQ.

Authors’ response: The ORCID iD is provided now.

Authors’ response to Reviewer

Dear Reviewer, thank you so much for your very constructive and crucial comments that help for the improvement of the manuscript.

1. Abstract: The result written in the abstract section is needed to re-write.

Authors’ Response: Thank you, dear reviewer, very much for comment. We have rewritten the result section.

2. Variable in study not been define clearly.

Authors’ Response: Thank you, dear reviewer, very much for the comment. Now, the study variables, response variable and independent variables are clearly and separately put. 

3. I am sceptical about the method used in the study; survival regression can be used in spite of Zero-inflated Poisson regression.

Authors’ Response: Thank you so much dear reviewer. In this study, the response/outcome variable was the number of children died before celebrating fifth birthday per woman in her lifetime. This ‘the number of children died before the age of 5 years’ can be zero if none of the children of mother died before the age of 5 years i.e, if mother didn’t lose under-five children in her lifetime, the number of the children died before the age of 5 years is considered zero. It can be 1 if a woman lost one child before turning age 5. It can also be 2, 3,…. As it is clearly known the response variable is count i.e, it can take on 0 or 1 or 2 or,…. In this study, the number of children died before age of 5 years ranged from 0 to 7. The appropriate models to analyse such count response data are the count regression models. For the excess zero outcome of the count data, Zero-inflated Poisson regression is appropriate. Based on the model comparison criteria, Zero-Inflated Poisson (ZIP) regression model was found to be the best model to fit the data. And, based on your comment, we modified the title as Zero-Inflated Poisson (ZIP) regression analysis of factors associated under-five mortality as the ZIP regression model best fitted the data compared to other count regression models used in the study.

4. Justifications of the methodology used has to be added.

Authors’ Response: Thank you, dear reviewer, so much. We added justification of the methodology used in the manuscript under section Statistical Data Analysis Methods 

5. What is newness in the study? and how it different from the existing studies form the same dataset.

Authors’ Response: Thank you, dear reviewer, for the question. Many of the studies previously done were analysed the data using logistic regression and survival regression analysis. But, this study was done analysing the data using the count regression analysis models. As the under-five mortality data experiences excess zero (0), the Zero-Inflated Poisson (ZIP) regression model was used. Moreover, this study included some additional important variables, such as type of cooking fuel, time to get source of water, age of household head, relationship to household head, number of under-five children, and number of children at home, which were not included in most of the previously studies from the same dataset. And, three new factors of under-five mortality, such as mothers’ time to get source of drinking water, household heads ages, and number of children at home, were found to be significant factors in this study as compared with previously conducted studies in Ethiopia.

---

## [Decision Letter · Decision Letter 1]

30 Aug 2023

Zero-inflated poisson regression analysis of factors associated with under-five mortality in Ethiopia using 2019 Ethiopian demographic and health survey data

PONE-D-22-20230R1

Dear Dr. Argawu,

We’re pleased to inform you that your manuscript has been judged scientifically suitable for publication and will be formally accepted for publication once it meets all outstanding technical requirements.

Kind regards,

Aklilu Habte, MPH

Academic Editor

PLOS ONE

Additional Editor Comments (optional):

Comments from the reviewer, Pushpendra Singh, PhD

All comments have been addressed

Thank you, authors, for your response to my comments and I appreciate the revisions you have made based on my suggestions.

Final decision: Accept

Reviewers' comments:

Reviewer's Responses to Questions

**Comments to the Author**

1. If the authors have adequately addressed your comments raised in a previous round of review and you feel that this manuscript is now acceptable for publication, you may indicate that here to bypass the “Comments to the Author” section, enter your conflict of interest statement in the “Confidential to Editor” section, and submit your "Accept" recommendation.

Reviewer #1: All comments have been addressed

2. Is the manuscript technically sound, and do the data support the conclusions?

Reviewer #1: Yes

3. Has the statistical analysis been performed appropriately and rigorously? 

Reviewer #1: Yes

4. Have the authors made all data underlying the findings in their manuscript fully available?

Reviewer #1: (No Response)

5. Is the manuscript presented in an intelligible fashion and written in standard English?

Reviewer #1: Yes

6. Review Comments to the Author

Reviewer #1: Thank you, authors, for your response to my comments. I appreciate the revisions you have made based on my suggestions.

7. PLOS authors have the option to publish the peer review history of their article (what does this mean?). If published, this will include your full peer review and any attached files.

Reviewer #1: No

---

## [Editor Report · Acceptance letter]

3 Nov 2023

PONE-D-22-20230R1 

Zero-inflated Poisson regression analysis of factors associated with under-five mortality in Ethiopia using 2019 Ethiopian mini demographic and health survey data 

Dear Dr. Argawu:

I'm pleased to inform you that your manuscript has been deemed suitable for publication in PLOS ONE. Congratulations! Your manuscript is now with our production department. 

Kind regards, 

on behalf of

Dr. Aklilu Habte 

Academic Editor

PLOS ONE